# Progress on the Synthesis of the Aromathecin Family of Compounds: An Overview

**DOI:** 10.3390/molecules29102380

**Published:** 2024-05-18

**Authors:** Takashi Nishiyama, Shota Mizuno, Yuhzo Hieda, Tominari Choshi

**Affiliations:** Faculty of Pharmacy and Pharmaceutical Sciences, Fukuyama University, 1 Sanzo, Gakuen-cho, Fukuyama 729-0292, Japan; t_nishiyama@fukuyama-u.ac.jp (T.N.); s.mizuno@fukuyama-u.ac.jp (S.M.); hieda@fukuyama-u.ac.jp (Y.H.)

**Keywords:** aromathecin family, rosettacin, 22-hydroxyacuminatine, acuminatine

## Abstract

We present a systematic review of the methods developed for the synthesis of the aromathecin family of compounds (benz[6,7]indolizino[1,2-*b*]quinolin-11(13*H*)-ones) and their derivatives. These methods can be broadly classified into four categories based on the construction of pentacyclic structures: Category 1: by constructing a pyridone moiety (D-ring) on the pyrroloquinoline ring (A/B/C-ring), Category 2: by constructing a pyridine moiety (B-ring) on the pyrroloisoquinolone ring (C/D/E-ring), Category 3: by constructing an indolizidinone moiety (C/D-ring) in a tandem reaction, and Category 4: by constructing a pyrrolidine moiety (C-ring) on the isoquinolone ring (D/E-ring).

## 1. Introduction

The indolizidine and quinolizidine moieties, which form the central core of indolizidine and quinolizidine alkaloids, are widely distributed in nature. It has been reported in the literature that natural products containing indolizidine and quinolizidine moieties in their structures exhibit various biological activities [1,2,3]. As a result, chemical compounds containing indolizidine or quinolizidine moieties have attracted significant attention from many medicinal and organic chemists. An indolizidine ring features a five-membered ring fused with a six-membered ring; these two rings share a nitrogen atom. A quinolizidine ring features two fused six-membered rings that share a nitrogen atom. These two heterocycles are often used together in structure–activity relationship studies of chemical compounds that contain them. This is because extending the five-membered ring moiety of indolizidine to a six-membered ring results in the formation of quinolizidine. Figure 1A shows the camptothecins (**1**–**4**), aromathecin family (**5**–**7**), and 8-oxoprotoberberine (**8**), which contain indolizidine and quinolizidine in their core structures.

Camptothecin (**1**) was isolated by Wani et al. [1] from the Chinese tree *Camptotheca acuminata* in 1966. Camptothecin features a pentacyclic structure where an indolizidin-2-one ring (C/D-ring) is fused with a quinoline (A/B-ring) and a lactone ring (E-ring). It has been shown that this alkaloid potently inhibits tumor growth by binding to the topoisomerase I enzyme (Top1). Subsequently, drug development studies were conducted where **1** was the lead compound, and irinotecan (**2**), topotecan (**3**), and belotecan (**4**) with substituents on the A/B-ring were developed for clinical studies. However, the hydrolysis of the E-ring lactone moiety produces hydroxycarboxylates with a high affinity for the human serum albumin protein; thus, the E-ring lactone hydrolysis product is responsible for reducing the activity of the derivatives of **1** [4]. To address this issue, the development of novel anticancer drugs has focused on the aromathecin family of compounds (benz[6,7]indolizino[1,2-*b*]quinolin-11(13*H*)-ones) where the lactone moiety of **1** is replaced by a benzene ring. To this day, three members of the aromathecin family are known: rosettacin (**5**), 22-hydroxyacuminatine (**6**), and acuminatine (**7**). **5** has been synthesized during the investigation of the application scope of novel synthetic method for **1**, and has been first reported as benz[6,7]indolizino[1,2-*b*]quinolin-11(13*H*)-one in 1969 [5]. It was later named “rosettacin” (**5**) by Hecht et al. [6]. Together with **1**, **6** has been isolated in a very low yield from *C. accuminata* seeds [7]. Furthermore, it has been reported that rosettacin, its derivatives, and **6** exhibit weak Top1 inhibitory activity. Therefore, aromathecins can be considered as a new class of Top1 inhibitors that can replace camptothecins in the development of therapeutics. To this day, many approaches to synthesizing aromathecins have been reported.

Additionally, protoberberine alkaloids and their derivatives exhibit a broad range of biological activities and have been predominantly used as active components in many traditional medicines, especially in China and other Asian countries [8].

In particular, 8-oxoprotoberberine alkaloids, such as oxypalmatine (**8**) [9], are tetracyclic compounds containing quinolizidin-2-one, which carries a carbonyl group in the quinolizidine moiety of protoberberine. It has been reported that these compounds feature unique structures and exhibit a wide range of therapeutic properties, including antitumor activity.

It is anticipated that their biological activities have antibacterial, antidiabetic, anticancer, and antihypertensive effects; as a result, they have attracted significant attention. Many approaches for synthesizing biologically active protoberberine alkaloids have been reported in the literature.

Furthermore, aromathecin and 8-oxoprotoberberine alkaloids feature similar structures and both exhibit anticancer activity. It is highly desirable to develop an efficient strategy for synthesizing both aromathecins and 8-oxoprotoberberines and to conduct structure–activity relationship studies.

Additionally, Figure 1B shows other typical alkaloids containing these two heterocycles in their core structure. Many indolizidine alkaloids with various substituents on a simple indolizidine ring have been reported in the literature. Since the structure of these natural products consists of several chiral centers, their structural characteristics and biological activities have attracted the attention of the research community, and asymmetric synthesis studies have been conducted. Details have been reported by Michael [2].

The indolizidine alkaloid septicine (**9**), which carries an aryl group at the 3- and 4-positions of indolizine, was isolated by Russel [10] from *Ficus septica*, which is a plant belonging to the Moraceae family; this plant is considered a biogenetic precursor to the phenanthroindolizidine alkaloid tylophorine (**10**) [11].

It has been reported that indolizidine alkaloids, such as septicine, exhibit anti-inflammatory [12] and anticancer activities [13]. On the other hand, phenanthroindolizidine alkaloids, such as tylophorine (**10**), are pentacyclic compounds where a phenanthrene ring is fused with an indolizidine ring.

Furthermore, pentacyclic compounds, such as cryptopleurine (**11**) [14], are referred to as phenanthroquinolizidine alkaloids where the indolizidine moiety of phenanthroindolizidine is replaced by quinolizidine.

Phenanthroizidine alkaloids (phenanthroindolizidine and phenanthroquinolizidine), such as tylophorine (**10**) and cryptopleurine (**11**), are a group of plant-derived compounds that can be potentially used as therapeutic agents. These plant extracts have been used in herbal medicine, and isolated compounds exhibit various promising therapeutic properties such as anti-ameobicidal [15], antiviral [16], anti-inflammatory [17] and anticancer activities [18]. Additionally, they are considered important target molecules by both synthetic and medicinal chemists, and numerous synthetic strategies have been developed to further investigate their biological activities. Details have been reported by Huang et al. [3].

As mentioned above, various synthetic strategies of natural products, including indolizidine and quinolizidine, have been reported, and significant research on drugs that can be used in the development of medicines has been conducted.

In this review, we present various strategies that have been developed to this day for the total synthesis of the aromathecin family of compounds, including the 8-oxoprotoberberine synthesis. To simplify the discussion, these strategies are broadly classified into four categories based on their synthetic routes and precursors.

A pentacyclic structure can be constructed using the following methods:Category 1: by constructing a pyridone moiety (D-ring) on the pyrroloquinoline ring (A/B/C-ring).Category 2: by constructing a pyridine moiety (B-ring) on the pyrroloisoquinolone ring (C/D/E-ring)Category 3: by constructing an indolizidinone moiety (C/D-ring) in a tandem reaction.Category 4: by constructing a pyrrolidine moiety (C-ring) on the isoquinolone ring (D/E-ring).

## 2. Review of Synthetic Methods

### 2.1. Category 1

Camptothecin (CPT: **1**) was isolated by Wani et al. [1] in 1966. Its binding with Top1 showed that it can potently inhibit tumor growth. Subsequently, drug development studies were conducted considering CPT as the lead compound, and various efficient methods for synthesizing CPT were reported. Among them, a method for synthesizing aromathecins was developed to investigate the scope of application of this method.

In this category of synthetic methods, various strategies for synthesizing a pyrroloquinoline ring (A/B/C-ring) have been reported. A pyridone ring (D-ring) is formed to construct a pentacyclic structure. Three synthetic routes are presented below.

In 1980, Pandit et al. [19] developed a synthetic route for **1** using pyrroloquinoline as the key compound and reported the total synthesis of **5** as an application of their method. Initially, the key precursor pyrroloquinoline (**15**) was synthesized using the method shown in Figure 1 [20]. By heating 3-oxopyrrolidine (**12**) and 2-aminobenzaldehyde (**13**) together with *p*-toluenesulfonic acid (*p*TSA) at 190–195 °C, the Friedlander reaction occurred, and *N*-(ethoxycarbonyl)pyrroloquinoline (**14**) was obtained with an 88% yield (Figure 1). To remove the ethoxycarbonyl group of **14**, **14** was treated in 48% hydrobromic acid (HBr) under reflux conditions to obtain the desired pyrroloquinoline (**15**) as a stable dihydrobromide salt [21]. Before its use, **15**·2HBr was treated using Et_3_N to obtain a free **15**, and the following reaction was performed without purification. β-Formylamide (**17**) was obtained from the reaction of **15** and 3-acetoxyphthalide (**16**) in the presence of potassium acetate (KOAc) in toluene at 80 °C with a 41% yield. Subsequently, when the amide (**17**) reacted with KOAc in acetic acid at 60 °C, an intramolecular cyclization reaction occurred, and a pyridone ring was formed, resulting in the total synthesis of **5** with a 76% yield.

In 2003, Cushman et al. [22] synthesized **5** and 8,9-dimethoxyrosettacin (**20**) as a derivative to study the structure–activity relationship of tetracyclic indenoisoquinoline derivatives. Additionally, they evaluated the cytotoxicity and Top1 inhibitory activity of these two compounds and reported their results.

As shown in Figure 2, dimesylate (**18**) [23,24] was treated using liquid NH_3_ in tetrahydrofuran (THF) at room temperature (rt) to obtain pyrroloquinoline (**15**); then, the excess NH_3_ was removed. Et_3_N and THF were added to the resulting solution of intermediate **15**; after stirring for 30 min at rt, bromide (**19a**) or chloride (**19b**) was added, and the mixture was further stirred to obtain **20** and **5** with 53% and 55% yields (in three steps), respectively.

In 2016, Gao et al. [25] proposed a flexible strategy for the efficient synthesis of CPT based on a cascade cyclization reaction for the construction of the indolizidinone scaffold. Additionally, they presented the total synthesis of rosettacin and 22-hydroxyacuminatine.

3-Aminomethyl-2-ethynylquinoline (**26**) was synthesized from 2-chloroquinoline-3-carbaldehyde (**21**) in six steps (Figure 3A). A reduction of **21** using NaBH_4_ in THF followed by an alcohol treatment using diphenylphosphoryl azide (DPPA) in the presence of 1,8-diazabicyclo[5.4.0]undec-7-ene (DBU) in THF was performed to obtain the azide (**22**). **22** was treated using triphenylphosphine (PPh_3_) to convert the azido group into an amino group; then, the amine was treated using Boc_2_O (Boc: *tert*-butyloxycarbonyl) to obtain the *N*-Boc-amine (**23**) form with a 78% yield (in four steps). The Sonogashira reaction of **23** with tetramethylsilyl (TMS)-acetylene (**24**) in the presence of bis(triphenylphosphine)palladium(II) chloride (PdCl_2_(PPh_3_)_2_), triethylamine (Et_3_N), and CuI produced 2-TMS-ethynylquinoline (**25**); then, 2-ethynylquinoline (**26**) was obtained with an 88% yield (in two steps) by treating **25** using K_2_CO_3_ in methanol (MeOH) to remove the TMS group.

Next, as shown in Figure 3B, the Sonogashira reaction of **26** with triflate (**27a,b**) in the presence of PdCl_2_(PPh_3_)_2_, Et_3_N, and CuI produced 2-arylethynylquinolines (**28a**,**29**) with 70% and 89% yields, respectively. Furthermore, **29** was treated using I_2_ and potassium hydroxide (KOH) in MeOH to convert its formyl group into a methoxycarbonyl group, and the methyl ester (**28b**) with a 70% yield was produced.

As shown in Figure 3C, the obtained **28a,b** was treated using trifluoroacetic acid (TFA) in CH_2_Cl_2_ to remove its Boc group; then, the resulting crude amine reacted with Cs_2_CO_3_ in MeOH. Exo-type hydroamination initially occurred between the alkyne moiety and the amino group, and intermediate pyrroloquinolines (**30a,b**) were produced; then, spontaneous lactamization resulted in the construction of the indolizidine moiety. As a result, **5** with a 90% yield was obtained (in two steps), and its derivative (**31**) was synthesized. Furthermore, **31** was treated using 2N HCl in MeOH to obtain **6** with a 70% yield (in three steps). They also reported the synthesis of several aromathecin derivatives as an application of this synthetic route. In addition, they reported the total synthesis of the 8-oxoprotoberberine alkaloid oxypalmatime and its derivatives using this synthetic route.

### 2.2. Category 2

In this category, various methods for synthesizing a 2,3-dihydropyrrolo[1,2-*b*]isoquinolin-1,5-dione structure have been reported. A pentacyclic structure can be synthesized by constructing a quinoline ring (A/B-ring) through the Friedlander reaction between pyrroloisoquinolin-1,5-dione (C/D/E-ring) and 2-aminobenzaldehyde derivatives. Five synthetic routes are presented below.

In 1969, Shamma and Novak [5] developed a method for synthesizing tricyclic pyrroloisoquinoline-1,5-dione using pyrrolidine-3-one ethylene ketal and reported the synthesis of rosettacin.

As shown in Figure 4, the pyrrolidin-3-one ethylene ketal (**32**) was prepared from pyrrolidin-3-one and treated using methyl 2-(chlorocarbonyl)benzoate (**33**) and K_2_CO_3_ to produce the amide (**34**) with an 89% yield. Next, a reduction of **34** using NaBH_4_ was performed to obtain an alcohol (**35**) with a 69% yield, which was then oxidized using MnO_2_ to produce an aldehyde (**36**) with a 92% yield. Subsequently, when **36** was treated in conc.H_2_SO_4_ at rt, the deketalization and cyclization proceeded in one step, producing pyrroloisoquinoline-1,5-dione (**37**) with a 55% yield. Finally, the Friedlander reaction of **37** and 2-aminobenzaldehyde (**13**) with Triton B in EtOH produced **5** with a 62% yield.

In 2006, Kanazawa et al. [26] reported the synthesis of 22-hydroxyacuminatine using a flash vacuum pyrolytic (FVP) cyclization and the Friedlander reaction as key reactions.

Initially, hydroxypyridone (**38**) was treated using Tf_2_O in pyridine to obtain a triflate (**39**) with an 89% yield, followed by the Heck reaction of **39** with methyl 2,4-pentadienoate (**40**) in the presence of PdCl_2_(PPh_3_)_2_ and Et_3_N to obtain an alkene (**41**) with a 64% yield (Figure 5). Next, **41** was subjected to FVP cyclization to obtain pyrroloisoquinolin-5-one (**42**) with a 77% yield; to introduce a carbonyl group to the 1-position, a ketone (**43**) with a 75% yield was obtained through SeO_2_ and Dess–Martin periodinane (DMP) oxidations. The Friedlander reaction of **43** and *N*-(2-aminobenzylidene)-*p*-toluidine (**44**) with *p*TSA in toluene produced methoxycarbonyl-rosettacin (**45**) with a 95% yield, which was used to construct a quinoline moiety. Finally, a reduction in the ester moiety of **45** using diisobutylaluminium (DIBAL) hydride in CH_2_Cl_2_ was performed to obtain **6** with a 70% yield. Furthermore, the 22-hydroxyacuminatine derivatives synthesized by using this method and evaluated for antiproliferative activity on cancer cell lines and for Top1 inhibitory activity [27].

In 2008, Daïch et al. [28] developed an efficient method for constructing pyrroloisoquinolinone rings where a domino *N*-amide acylation/aldol-type condensation occurred between a hydroxybenzotriazole (HOBt) ester and a lactam in the presence of NaH. The HOBt ester (**48**) was prepared in a quantitative yield (QY) by treating a 2-(methoxycarbonylmethyl)benzoic acid (**46**) and 1-hydroxybenzotriazole (**47**) with dicyclohexylcarbodiimide (DCC) in THF (Figure 6). The obtained **48** and **49** reacted with NaH to obtain pyrroloisoquinolone (**50**) with an 82% yield through a key reaction where *N*-acylation of the amide followed by intramolecular cyclization proceeded continuously. **50** was treated using Brederek’s reagent, followed by NaIO_4_, and the 1-position of **50** was oxidized to produce pyrroloisoquinoline-1,5-dione (**51**) with a 60% yield. The Friedlander reaction of **51** and **13** with *p*TSA in toluene produced 6-methoxycarbonyl-rosettacin (**52**) with a 51% yield. Finally, **52** in 48% HBr was heated at 135 °C to decarboxylate the methyl ester moiety, and **5** with a 61% yield was produced.

In 2012, Park et al. [29] reported a synthesis method for rosettacin; they developed an efficient and practical rhodium(III)-catalyzed intramolecular annulation of alkyne-tethered hydroxamic esters for the synthesis of 3-hydroxyalkyl isoquinolones as a key reaction.

*N*-((5-(Trimethylsilyl)pent-4-yn-1-yl)oxy)benzamide (**57**) was synthesized using the method shown in Figure 7. Pent-4-yn-1-ol (**53**) and *N*-hydroxyphthalimide (**54**) were treated using PPh_3_ and diisopropyl azodicarboxylate (DIAD); subsequently, they were subjected to the Mitsunobu reaction to obtain 2-((but-4-yn-1-yl)oxy)isoindoline-1,3-dione (**55**) with a 91% yield. Then, **55** reacted with trimethylsilyl trifluoromethanesulfonate (TMSOTf) in the presence of Zn(OTf)_2_ and Et_3_N in CH_2_Cl_2_ to obtain 2-((5-(trimethylsilyl)pent-4-yn-1-yl)oxy)isoindoline-1,3-dione (**56**) with a 97% yield. Next, **56** was treated using hydrazine to obtain *O*-alkoxylamine and then benzoylated to obtain **57** with a 92% yield. **57** was treated using (Cp*RhCl_2_)_2_ and CsOAc in MeOH at 60 °C to obtain isoquinolone (**58**) with a 98% yield. Subsequently, **58** was treated using DIAD and PPh_3_ to construct a pyrrolidine ring using intramolecular Mitsunobu-type cyclization; then, the TMS group was removed using tetra-*n*-butylammonium fluoride (TBAF) treatment to obtain pyrroloisoquinolone (**59**) with an 89% yield. Subsequently, **59** was subjected to SeO_2_ and DMP oxidations to obtain a ketone (**37**) with an 83% yield. Finally, **5** with a 94% yield was obtained from the reaction of **37** with **44** under Friedlander reaction conditions to construct the quinoline moiety.

In 2017, Glorius et al. [30] developed an efficient and highly regioselective synthetic method for isoquinolones by employing a Cp*Co^III^-catalyzed intramolecular C–H activation approach as a key reaction and achieved the total synthesis of rosettacin.

The key precursor *N*-((pent-4-yn-1-yl)oxy)benzamide (**61**) was synthesized using a similar synthetic route, as shown in Figure 8. Initially, 2-((but-4-yn-1-yl)oxy)isoindoline-1,3-dione reacted with hydrazine to obtain (pent-4-yn-1-yl)oxyamine (**60**); then, **60** and benzoyl chloride reacted with K_2_CO_3_ in EtOAc-H_2_O to obtain the desired **61** with a 64% yield. **61** was treated with Cp*Co(CO)I_2_, AgSbF_6_, CsOPiv, and PivOH in 2,2,2-trifluoroethanol (TFE) at 100 °C to obtain isoquinolone (**62**) with an 86% yield. A Mitsunobu reaction of the isoquinolone **62** with DIAD and PPh_3_ in THF produced pyrroloisoquinolone (**59**) with a 93% yield. Subsequently, **59** was subjected to a sequential oxidation using SeO_2_/PCC to prepare a ketone (**37**). **37** and **44** were treated using *p*TSA in toluene at 130 °C to obtain **5** with a 95% yield.

### 2.3. Category 3

Transition-metal-catalyzed cyclization reactions via C-H activation have been widely used as an efficient method for constructing complex molecules. In particular, regarding the insertion of alkynes into aromatic compounds with different directing groups, a direct aryl C-H functionalization using these catalysts is a widely investigated reaction, and its usefulness in synthesizing diverse heterocycles has been demonstrated.

In this category of synthetic methods, the key precursors used are synthesized by appropriately combining quinoline (A/B-ring), benzamide (E-ring), and alkynes. Then, this compound reacts in the presence of a transition metal catalyst, and an indolizidinone ring (C/D-ring) is constructed through a cascade reaction. As a result, a pentacyclic structure is constructed. Five synthesis routes are presented below.

Eycken et al. [31,32,33] developed and reported three rhodium(III)-catalyzed synthetic methods for constructing fused indolizidinone moieties with an aromatic or a heteroaromatic ring. By applying these methods, they synthesized rosettacin.

In the first method [31] of the synthesis of fused indolizidinone (2017), an efficient rhodium(III)-catalyzed intramolecular annulation was developed for benzamides, which were substituted by an aryl group carrying an alkynyl group on the nitrogen. The synthesis of the key isoquinolone synthesis precursors *N*-((2-trimethylsilylethynylquinolin-3-yl)methyl)benzamide (**65**) was based on the method shown in Figure 9. The Sonogashira reaction of 2-chloroquinoline (**23**) with *tert*-butyldimethylsilyl (TBS)-acetylene (**63**) in the presence of PdCl_2_(PPh_3_)_2_, CuI, and Et_3_N in toluene produced 2-ethynylquinoline (**64**). **64** was treated with TFA in CH_2_Cl_2_ to remove the Boc group, followed by acylation with benzoyl chloride to obtain the desired **65** with a 50% yield (in three steps). Next, **65** was treated with (Cp*RhCl_2_)_2_, Cu(OAc)_2_, and CsOAc in *t*-AmOH at 110 °C to obtain 6-TBS-rosettacin (**66**) with a 71% yield. Subsequently, **66** reacted with TBAF to remove the TBS group, and **5** was obtained with an 88% yield. In addition, they reported the total synthesis of the 8-oxoprotoberberine alkaloid oxypalmatime using this synthetic route.

In the second method (2018) [32], an intermolecular cascade annulation of 2-acetylenic aldehydes with *O*-substituted *N*-hydroxybenzamides through a rhodium(III)-catalyzed C–H activation was developed for the construction of hydroxyindolizidinone moieties.

As shown in Figure 10, the Sonogashira reaction of 2-chloroquinoline-3-carbaldehyde (**21**) and TBS-acetylene (**63**) with PdCl_2_(PPh_3_)_2_ and CuI in Et_3_N produced 2-ethynylquinoline (**67**). **67** and *N*-pivaloyloxybenzamide (**68**) were treated using (Cp*RhCl_2_)_2_, CsOAc, and PivOH in MeCN at 60 °C to obtain 6-TBS-rosettacin (**69**) with a 65% yield. Subsequently, **69** was treated with TBAF to remove the TBS group, followed by treatment using BF_3_·Et_2_O/Et_3_SiH to produce **5** with a 42% yield in a one-pot synthesis.

In the third method (2019) [33], a novel intramolecular cascade annulation of *O*-substituted *N*-hydroxybenzamides was developed for the synthesis of indolizidinones through a rhodium(III)-catalyzed sequential C(sp2)-H activation and a C(sp3)-H amination.

The synthesis of the key isoquinolone synthesis precursor hydroxamic ester (**74**) was based on the method shown in Figure 11. A reduction of 2-ethynylquinoline-3-carbaldehyde (**67**) using NaBH_4_ in THF-H_2_O was performed to obtain the alcohol (**70**). Then, **70** was treated using CBr_4_ and PPh_3_ in CH_2_Cl_2_ to convert the hydroxy group into a bromo group, and *N*-hydroxyphthalimide reacted with DBU to produce *N*-alkyloxyphthalimide (**72**). Next, **72** was treated using hydrazine in MeOH-CH_2_Cl_2_ to obtain *O*-alkylhydroxylamine (**73**). The acylation of **73** and benzoyl chloride with K_2_CO_3_ in EtOAc-H_2_O produced the desired **74**. Then, **74** reacted with (Cp*RhCl_2_)_2_ and CsOAc in 1,4-dioxane at 60 °C under air to produce 13-hydroxy-6-TBS-rosettacin (**69**) with a 46% yield. Subsequently, **69** was treated with TBAF to remove the TBS group, followed by a reduction of the resulting 13-hydroxyrosettacin under BF_3_·Et_2_O/Et_3_SiH conditions to obtain **5** with a 46% yield (in two steps).

In 2018, Evano et al. [34] developed a copper-catalyzed photoinduced radical domino cyclization reaction of ynamides. In this process, a radical intermediate was generated from the C-I bond using a copper catalyst, and a 5-exo-dig cyclization reaction occurred on the C≡C triple bond. As a result, a vinylic radical intermediate was obtained, which was then cyclized to obtain an arene through a 6-endo-trig process to construct an indolizine ring and, finally, aromatized to synthesize rosettacin.

The synthesis of the key isoquinolone precursor ynamide (**77**) was based on the method shown in Figure 12. Initially, the acylation of 3-aminomethyl-2-iodoquinoline (**75**) and benzoyl chloride using Et_3_N in CH_2_Cl_2_ was performed to obtain the benzamide, followed by treatment of the benzamide and [(trimethylsilyl)ethynyl]phenyliodonium triflate (**76**) using potassium bis(trimethylsilyl)amide (KHMDS) in toluene to obtain the desired ynamide (**77**) with a 40% yield (in two steps). Next, **77** reacted with [(DPEphos)(bcp)Cu]PF_6_, Cy_2_NiBu and K_2_CO_3_ in MeCN in a photoreactor under a 420 nm wavelength irradiation to produce 6-TMS-rosettacin (**78**) with a 71% yield. Finally, **78** was treated with TBAF to remove the TMS group, and **5** with an 88% yield was obtained.

In 2018, Reddy et al. [35] developed a one-pot method for the efficient synthesis of tetracyclic 7-hydroxyisoindolo[2,1-*b*]isoquinolin-5(7*H*)-one scaffold from *N*-(pivaloyloxy)amides and 2-alkynyl aldehydes through a rhodium(III)-catalyzed C−H functionalization.

Figure 13 shows an application of their proposed key reaction for synthesizing rosettacin. 2-Ethynylquinoline-3-carbaldehyde (**80**) with a quantitative yield (QY) was initially synthesized from 2-chloroquinoline-3-carbaldehyde (**21**) through the Sonogashira reaction and removal of the TMS group. **80** and *N*-pivaloyloxybenzamide (**68**) reacted with (Cp*RhCl_2_)_2_ and CsOAc in acetone to produce 13-hydroxyrosettacin (**81**) with a 66% yield. Finally, a reduction of the aminal moiety of **81** using BF_3_·Et_2_O/Et_3_SiH in CH_2_Cl_2_ was performed to obtain **5** with a 74% yield.

### 2.4. Category 4

In this category of synthetic methods, isoquinolone (D/E-ring) is synthesized to which quinoline (A/B-ring) is attached. The synthesis is performed by constructing a pentacyclic structure where a pyrrolidine ring (C-ring) is finally constructed. Three synthesis routes are presented below.

In 2015, Daïch et al. [36] achieved the synthesis of rosettacin using an *N*-alkylation lactam, followed by an aryl radical cyclization of enamides, which contained either a bromine or a chlorine atom as a radical precursor, as a key reaction.

As shown in Figure 14, the homophthalic acid (**82**) was treated using thionyl chloride (SOCl_2_) to obtain the homophthalic anhydride (**83**) with a 98% yield. Subsequently, **83** reacted with *N*,*N*-dimethylhydrazine in AcOH by employing reflux to produce an imide (**84**) with an 80% yield. After performing a reduction of **84** with NaBH_4_ in CH_2_Cl_2_–MeOH at 0 °C, 3M HCl was added to the reaction mixture and stirred at rt to obtain an enamide (**85**) with an 83% yield (in two steps). **85** was then treated using magnesium monoperoxyphthalate in MeOH to cleave the hydrazine moiety, and isoquinolone (**86**) with a 68% yield was obtained. **86** and 2-chloro-3-chloromethylquinoline (**87**) were treated using K_2_CO_3_, KI, and 18-crown-6 in toluene by employing reflux to obtain *N*-alkylated isoquinolone (**88**) with an 88% yield. Finally, by refluxing **88** in toluene using azobisisobutyronitrile (AIBN) and tris(trimethylsilyl)silane, a radical cyclization occurred, and the desired **5** with a 45% yield was synthesized.

In 2018, Huang et al. [37] reported a method for synthesizing rosettacin using an *N*-heterocyclic carbene (NHC)-catalyzed aerobic oxidation of an isoquinolinium salt to obtain isoquinolone as a key step. The isoquinolinium salt **91** was prepared using 2-bromo-3-bromomethylquinoline (**89**) and isoquinoline (**90**) in MeCN (Figure 15). **91** was treated using 2-mesityl-6,7-dihydro-5*H*-pyrrolo[2,1-*c*][1,2,4]triazolium tetrafluoroborate (**92**) and DBU in THF at −40 °C air conditions to obtain isoquinolone (**93**) with a 73% yield. Subsequently, **93** reacted in the presence of a Pd(OAc)_2_ catalyst, and a pyrrolidine ring was formed, and **5** with a 99% yield was synthesized. In addition, they reported the total synthesis of the 8-oxoprotoberberine alkaloid (±)-gusanlung D and ilicifoline using this synthetic route.

In 2023, Choshi et al. [38] developed a method for synthesizing the pentacyclic scaffold of the aromathecin family by constructing the indolizidinone moiety after the isoquinolone synthesis as a key step.

The synthesis of the key isoquinolone precursor 2-ethynylbenzaldehyde oxime was based on the method shown in Figure 16. 2-Iodo-3-hydroxymethylquinoline (**94**) and methyl iodide were treated using NaH to provide 2-iodo-3-methoxymethylquinoline (**95**) with an 86% yield. The Sonogashira reaction of **95** with 2-ethynylbenzaldehydes **96a,b** in the presence of PdCl_2_(PPh_3_)_2_, CuI, and Et_3_N produced benzaldehydes **97a,b** with 84% and 87% yields, respectively. By treating **97a,b** and hydroxylamine in the presence of AcONa, the desired oximes **98a,b** were obtained with 86% and 81% yields, respectively. Subsequently, oximes **98a,b** were heated in 1,2-dichlorobenzene (1,2-DCB) at 80 °C to obtain *N*-oxides **99a,b** with 73% and 26% yields, respectively. The Reissert–Henze-type reaction of **99a,b** in Ac_2_O at 50 °C under microwave irradiation produced isoquinolones **100a,b** with 73% and 52% yields, respectively. Finally, **100a** was treated using conc.H_2_SO_4_ in EtOH at 110 °C to obtain **5** with an 88% yield. However, although **100b** reacted under similar conditions, the desired 22-hydroxyacuminatine (**6**) was not obtained; instead, acuminatine (**11**) with a 79% yield was obtained. In addition, they reported the total synthesis of the 8-oxoprotoberberine alkaloid (alangiumkaloids A and B) using this synthetic route [39].

## 3. Conclusions

In this review, we provided a detailed description of all methods developed for synthesizing the aromathecin family of compounds (rosettacin, 22-hydroxyacuminatine, and acuminatine) and their derivatives. The methods described here are broadly classified into four categories based on the common intermediates or synthetic routes involved in the construction of the aromathecin skeletal structure. We anticipate that these methods will be used to efficiently synthesize derivatives, and new anticancer drugs will be developed based on these derivatives. We also anticipate that these methods will provide valuable information for developing novel and efficient synthetic routes.

## Data Availability

Not applicable.

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
