# Peer review of "Progress on the Synthesis of the Aromathecin Family of Compounds: An Overview"

_molecules, 2024, doi:10.3390/molecules29102380_

Round 1
Reviewer 1 Report
Comments and Suggestions for Authors
Please refer to the attachment.

Moderate editing of the English language is required. Some statements need to be revised.
Reviewer 2 Report
Comments and Suggestions for Authors
In the Abstract (lines 12-15) you have duplication of the numbers: "1) Category 1", "2) Category 2"!!! It will be better if it stands only ones, e.g. "Category 1, Category 2, etc.
In the Introduction you have duplication of almost the same sentence, linen 30-31 and lines 37-38. It will be nice if this sentence can be changed all skipped in lines 30-31!
Line 38: there are actually two aryl groups at the positions 3 and 4, not just one at position 3,4!!!
Line 216: number 25 should be without brackets!
Line 218: number 26 also should not be in brackets!
Line 581: "... reduction of 81 ""missing word"" NaBH4,..." , it should be "reduction of 81 with NaBH4" or reduction of 81 by NaBH4"!
Comments on the Quality of English LanguageLine 29: "...because, for example...." This sentence should be formulated better!
Reviewer 3 Report
Comments and Suggestions for Authors
This review article has presented a comprehensive view on the synthetic strategies for aromethacine moieties. The article is well written and worth publishing with my single concern that, in the introduction section, lines 72-73 need a reference here, for the anti-inflammatory and anticancer activities of Septicine, which is known as anti-biotic (or medicine for bacterial infections).
